# Absence of Genotype/Phenotype Correlations Requires Molecular Diagnostic to Ascertain Stargardt and Stargardt-Like Swiss Patients

**DOI:** 10.3390/genes12060812

**Published:** 2021-05-26

**Authors:** Virginie M.M. Buhler, Lieselotte Berger, André Schaller, Martin S. Zinkernagel, Sebastian Wolf, Pascal Escher

**Affiliations:** 1Department of Ophthalmology, Inselspital, Bern University Hospital, 3010 Bern, Switzerland; virginie.buehler@insel.ch (V.M.M.B.); lieselotteerika.berger@insel.ch (L.B.); martin.zinkernagel@insel.ch (M.S.Z.); sebastian.wolf@insel.ch (S.W.); 2Department of BioMedical Research, University of Bern, 3010 Bern, Switzerland; 3Department of Human Genetics, Inselspital, Bern University Hospital, 3010 Bern, Switzerland; andre.schaller@insel.ch

**Keywords:** macula dystrophy, retinal degeneration, choroidal dystrophy, Stargardt disease, central areolar choroidal dystrophy, genetic analyses, recessive inheritance, dominant inheritance

## Abstract

We genetically characterized 22 Swiss patients who had been diagnosed with Stargardt disease after clinical examination. We identified in 11 patients (50%) pathogenic bi-allelic *ABCA4* variants, c.1760+2T>C and c.4496T>C being novel. The dominantly inherited pathogenic *ELOVL4* c.810C>G p.(Tyr270*) and *PRPH2*-c.422A>G p.(Tyr141Cys) variants were identified in eight (36%) and three patients (14%), respectively. All patients harboring the *ELOVL4* c.810C>G p.(Tyr270*) variant originated from the same small Swiss area, identifying a founder mutation. In the *ABCA4* and *ELOVL4* cohorts, the clinical phenotypes of “flecks”, “atrophy”, and “bull’s eye like” were observed by fundus examination. In the small number of patients harboring the pathogenic *PRPH2* variant, we could observe both “flecks” and “atrophy” clinical phenotypes. The onset of disease, progression of visual acuity and clinical symptoms, inheritance patterns, fundus autofluorescence, and optical coherence tomography did not allow discrimination between the genetically heterogeneous Stargardt patients. The genetic heterogeneity observed in the relatively small Swiss population should prompt systematic genetic testing of clinically diagnosed Stargardt patients. The resulting molecular diagnostic is required to prevent potentially harmful vitamin A supplementation, to provide genetic counseling with respect to inheritance, and to schedule appropriate follow-up visits in the presence of increased risk of choroidal neovascularization.

## 1. Introduction

With a prevalence of 8–10/100,000, Stargardt disease is the most frequent inherited macular dystrophy (MD) in both children and adults [1]. The clinical spectrum of Stargardt disease is highly variable in terms of age of onset, phenotypical features, progression rate, and the extension of retinal involvement [2]. The first manifestations can already occur during the first two decades of life (juvenile-onset) [3], and the second peak of incidence is during early adulthood [1]. A significant number of patients become affected in late adulthood by the so-called “late-onset foveal sparing Stargardt disease” [4]. In terms of symptoms, the most often reported ones are visual impairment [2], bilateral central visual loss, dyschromatopsia, central scotoma [1], and more seldom photophobia or slow dark adaptation. The classical description of Stargardt disease relates to retinal flecks around the macula, variably extending to the mid-peripheral retina, and a progressive macular atrophy over time [2]. However, the disease’s phenotypical spectrum is much broader and includes: macular atrophy without flecks, bull’s-eye maculopathy-like, fundus flavimaculatus (flecks without atrophy), foveal-sparing phenotype, cone-rod dystrophy [5], and retinitis pigmentosa-like [6].

The classical form of Stargardt disease is caused by homozygous or compound heterozygous pathogenic variants in the adenosine triphosphate (ATP)-binding cassette (ABC) transporter 4 (*ABCA4*) gene. *ABCA4* is exclusively expressed in photoreceptors, where it is located on the outer segments disc membranes of rods and cones. Its function is the translocation of retinoids produced during the visual cycle, after the photobleaching-induced isomerization of 11-cis-retinal: N-retinylidene phosphatidylethanolamine and all-trans-retinal [7]. Pathogenic *ABCA4* variants, therefore, lead to the defective transport of retinoids [2]. Thus, the byproducts of the visual cycle (lipofuscin being the main one) accumulate in the photoreceptors’ outer segments and then in the retinal pigment epithelium (RPE) cells because of the disc shedding process [7]. This accumulation of lipofuscin and its highly toxic component A2E (N-retinylidene-N-retinylethanolamine) in RPE cells leads to their dysfunction and eventually to their death, followed by one of the photoreceptors [1], this substance having a very high epithelial and neuronal toxicity [7]. 

An inverse relationship between the residual ABCA4 function and the disease severity is consequently intuitive [4]. Nevertheless, genotype–phenotype correlations are limited among the more than 6000 pathogenic variants reported in the literature [8]. Missense variants, non-canonical splice site variants, and deep intronic variants are usually associated with a milder phenotype, even though they can be the origin of fast disease progression as well [9,10,11]. Nonsense and frameshift variants are typically responsible for more severe presentations, with an earlier vision loss and a worse prognosis, although in these cases as well, exceptions do occur [7]. Compound heterozygosity furthermore hampers genotype–phenotype correlations. For example, the most common pathogenic *ABCA4* variant c.5882G>A p.(Gly1961Glu) is linked to a mild disease in a homozygous state, but in a compound heterozygous state, milder to much more severe disease phenotypes are observed [8,12]

Similar clinical phenotypes that present in patients affected by *ABCA4*-linked autosomal recessive Stargardt disease (STGD1) can also be observed in *ELOVL4*-linked autosomal dominant Stargardt disease (STGD3) [13,14], in *PROM1*-linked autosomal dominant Stargardt disease (STGD4) [15], and in an autosomal dominant macular dystrophy associated with a c.422A>G p.(Tyr141Cys) *PRPH2* variant [16].

The ubiquitously expressed *ELOVL4* gene encodes the integral membrane fatty acid elongase ELOVL4, which elongates very long chain (VLC) saturated and polyunsaturated fatty acids (PUFA). In the retina, VLC-PUFA are enriched in the phosphatidylcholine fraction of retinal lipids and are tightly associated with opsins [17]. Two pathogenic frameshift variants, a 1-bp and a 5-bp deletion, and a c.810C>G, p.(Tyr270*) nonsense variant have been identified so far [14,18]. These truncated ELOVL4 proteins mislocalize to the Golgi or form perinuclear aggregates by physically interacting by a so-called dominant negative effect with wild-type ELOVL4, leading to cellular stress and eventually cell death [17,19].

The autosomal dominant Stargardt-like pattern dystrophy associated with the pathogenic *PRPH2* c.422A>G p.(Tyr141Cys) has some reported variability in clinical phenotypes, including butterfly-shaped pattern dystrophy, adult-onset foveomacular dystrophy, fundus flavimaculatus, and geographic atrophy [16,20]. *PRPH2* encodes the transmembrane tetraspanin glycoprotein Peripherin 2, previously known as RDS (retinal degeneration slow). Peripherin 2 is specifically located to the rim region of cone and rod outer segment discs, where it forms complexes with its non-glycosylated homolog ROM-1 (rod outer segment membrane 1) [21]. As peripherin-2 is required for the morphogenesis of the photoreceptors’ outer segments, its deficiency leads to cell disorganization and eventually apoptosis [22,23].

Given the genetic heterogeneity of patients presenting a Stargardt-like phenotype, we aimed at deciphering a genotype–phenotype correlation through the study of a cohort of Swiss patients.

## 2. Materials and Methods 

### 2.1. Patients

We included patients who had their initial visit or follow-up visits in the Department of Ophthalmology of the University Hospital of Bern (Inselspital, Bern, Switzerland) between 2018 and 2020. All patients provided written consent for genetic analyses and use of anonymized data for scientific purposes. In this retrospectively constituted cohort, we then included all the patients who had a confirmed genetic diagnosis of STGD1 (with ≥2 *ABCA4* pathogenic variants), STGD3 (all with the *ELOVL4* c.810C>G p.(Tyr270*) variant) and *PRPH2*-c.422A>G p.(Tyr141Cys)-associated autosomal dominant macular dystrophy.

### 2.2. Molecular Genetic Analysis

Two to five ml peripheral venous blood samples were collected in EDTA-containing tubes (S-Monovette^®^, Sarstedt, Nümbrecht, Germany). DNA was extracted from the EDTA blood using the Prepito DNA Blood 600 Kit (Qiagen, Hilden, Germany). The molecular genetic analysis was performed by exome sequencing using Trusight One normal or Trusight One expanded sequencing panels (Illumina) and next-generation sequencer MiSeq or NextSeq 500 (Illumina, San Diego, CA, USA). Sequence alignment and local realignment to the human reference genome (GRCh37hg19) were performed with Biomedical Genomics Workbench (Qiagen Bioinformatics, Hilden, Germany). Variants with an allele frequency <5% in the coding regions and the flanking intronic regions (± 8 bp) were evaluated. The following databases were used for data interpretation: dbSNP137, dbSNP150, 1000 Genomes, Exome variant server, NHLBI Exome Sequencing Project (ESP, ESP6500SI-V2), Genome Aggregation Database (gnomAD, v.2.0.2), ClinVar, HGMD professional, Intervar, VarSome, NCBI- and ENSEMBL databases. PCR amplification (Thermo Fisher Scientific, Waltham, MA, USA) and Sanger sequencing on an ABI Prism 3500XL Genetic Analyzer (Applied Biosystems, Foster City, CA, USA) confirmed pathogenic and likely pathogenic sequence variants. In case of very high clinical suspicion of a particular mutation, or in case of analysis of the relatives of a patient who already got a genetic diagnosis, DNA was extracted from the mouth mucosal cell swabs (FAB-SWAB, Loci Forensic Products, Nieuw-Vennep, The Netherlands) via Qiagen-Kit Mini and the relevant exons directly analyzed by Sanger sequencing.

### 2.3. Clinical Data

We performed a medical record review on the patients. The collected data included the patients’ sex and age, medical history (general and ophthalmological, with a special focus on the age at the first symptoms presentation and on the described ophthalmological symptoms), best corrected visual acuity and its evolution, and finally, family history. For every patient, we analyzed spectral domain optic coherence tomography (SD-OCT) images and autofluorescence pictures. These images were acquired using the Heidelberg Spectralis HRA+OCT device (Heidelberg Engineering, Heidelberg, Germany) and interpreted independently by three of the authors. 

## 3. Results

The *ABCA4* cohort is composed of 11 unrelated patients, 9 females and 2 males. Their family trees are drawn in Figure 1 and their genotypes described in Table 1. Genetic analyses identified compound heterozygosity in all patients, including two novel *ABCA4* variants: a splice variant c.1760+2T>C p.? and a missense variant c.4496T>C p.(Leu1499Pro). The first variant is classified as pathogenic and the latter as likely pathogenic according to the criteria of the American College of Medical Genetics and Genomics (ACMG) [24]. In patient II.1 of family 7, we identified the frequent complex allele p.[(Leu541Pro;Ala1038Val)]. Compound-heterozygosity of patient III.1 of the consanguineous family 11, originating initially from the Caucasus region, was associated with a pseudo-dominant inheritance pattern. The affected mother was homozygous for the likely pathogenic c.1807T>C p.(Tyr603His) variant, whereas her son was compound heterozygous for the paternally inherited c.4496T>C p.(Leu1499Pro) and the maternally inherited c.1807T>C p.(Tyr603His) variants. We also checked for the presence of the frequent hypomorphic variant c.5603A>T p.(Asn1868Ile) and identified it in patient I.2 of family 10.

The median age of the patients at the time of the study (2020) was 44 years, ranging from 18 to 69 years old. The median age at the beginning of the symptoms was 35.5 years (range: 10–57 years old; unknown for a patient). The average duration of the disease, measured by the time interval between the first symptoms and the year this study was conducted was 8 years (range: 2–17 years). 

The visual acuity was considered as good (≥0.5) at the final examination in 7 eyes among the 10 patients for which we had a visual acuity value (35%). It was considered as normal (≥0.8) [4] in 6 eyes (30%). The values and the evaluations of the visual acuities related to the patients’ ages are represented in Figure 2 and illustrate the typical “juvenile-onset” and “late-onset” Stargardt patients of this *ABCA4* cohort.

Out of these 11 patients, we were able to obtain clinical data from 9 of them. The most common complaints were a progressive vision decrease (8/9; 88.89%), followed by hemeralopia and photophobia (4/9; 44.44%). More seldom, patients would spontaneously relate about reading difficulties (3/9; 33.33%), color vision deterioration (1/9; 11.11%), or about the presence of a central scotoma (1/9; 11.11%). The fundus autofluorescence and OCT data are respectively shown in Figure 3 and Figure 4, and the findings are summarized in Table 2.

Eight patients (4 females and 4 males) were part of the *ELOVL4* cohort. Most of them were related, as shown in their family trees in Figure 5. All of them originated from the same alpine region located in the South of the Canton of Valais. They all shared the same heterozygous pathogenic c.810C>G p.(Tyr270*) variant in the *ELOVL4* gene, indicating a founder mutation (see Table 1).

When this study was conducted in 2020, the median age of these patients was 35.5 years old (range: 19–82 years old). Their median age when they first became symptomatic was 27 years old, with a very broad range going from around 10 to 79 years old. One patient was still asymptomatic at his last ophthalmological control at age 23 (family 13, patient III.1). The median duration of the disease for the 7 symptomatic patients (determined with the same method as the one described above) was 4 years (range: 1–17 years).

Ten eyes from these 8 patients (62.5%) had a good visual acuity (≥0.5) at their last ophthalmological check-up, whereas 7 out of them (43.8%) could even be considered as having a normal vision (visual acuity ≥ 0.8). The evolution of their visual acuity is shown in Figure 6. While analyzing the reported symptoms of the 7 symptomatic *ELOVL4* patients, we noticed that all of them (100%) were complaining about a progressive vision decrease. The other reported symptoms were less common but similar to the ones of the *ABCA4* cohort: reading difficulties (1/7, 14.29%), hemeralopia (1/7, 14.29%), photophobia (2/7, 28.57%), color vision deterioration (1/7, 14.29%), and central scotoma (1/7, 14.29%). The fundus autofluorescence and OCT data are respectively shown in Figure 7 and Figure 8, and the findings are summarized in Table 2.

Our cohort with patients harboring the pathogenic c.422A>G, p.(Tyr141Cys) *PRPH2* variant was the smallest one and counted only 3 unrelated patients, 2 females and 1 male. Their family trees are shown in Figure 9. They all originated from a region located between the cantons of Fribourg and Bern, indicating a possible founder effect.

In 2020, their ages were in chronological order 39, 51, and 69 years old and their symptoms first started when they respectively were 7, 10, and 36 years old, which gave a disease duration of 32, 41, and 33 years, which is much longer than what we had in the two previous cohorts.

Five eyes from these three patients (83.33%) had at their last ophthalmological control a visual acuity considered as good (≥0.5) included three (50%) who had one considered as normal (≥0.8). Their vision evolution is schematized in Figure 10. All of them (100%) were complaining about a progressive diminution of their vision. Only one patient (33.33%) reported reading difficulties, one (33.33%) hemeralopia, and one (33.33%) a central scotoma. However, none of them related about photophobia or color vision disturbances. The fundus autofluorescence and OCT data are respectively shown in Figure 11 and Figure 12, and the findings are summarized in Table 2.

While analyzing the fundus autofluorescence images of all patients, irrespective of their genotype, we identified three main phenotypes that we respectively named “flecks”, “atrophy”, and “bull’s eye like”.

The “flecks” phenotype is the most prevalent one among our three cohorts: 7/11 = 63.64% in the *ABCA4* cohort, 3/6 = 50% in the *ELOVL4* cohort and 2/3 = 66.67% in the *PRPH2* cohort. In the “flecks” phenotype, a central mottled pattern of hyperautofluorescent and hypoautofluorescent flecks variably extend to the mid-periphery, and we observed an almost identical phenotype in *ABCA4*, *ELOVL4*, and *PRPH2* patients (Figure 13). We observed small differences among the patients exhibiting a “flecks” phenotype. For example, in patient 5/II.2, this phenotype was only present in the periphery and absent in the macula, which makes it atypical for a macular dystrophy. Some other patients presented a higher number of hyperautofluorescent flecks and a smaller mottled pattern (7/II.1 and 8/III.2) and some other ones a pattern looking more like a butterfly-like pattern than a mottled one (17/II.1 and 18/II.2).

The next phenotype we recognized was the “atrophy” one, characterized by a variable extent of macular atrophy. Again, we observed similar phenotypes in *ABCA4*, *ELOVL4,* and *PRPH2* patients (Figure 14). Some other patients presented a form of nummular atrophy combined with the “flecks” phenotype (patient 13/I.1 and 13/II.2).

The last phenotype that we recognized in our case series was the “bull’s eye like” one (Figure 15). We identified this phenotype only in patients of the *ABCA4* and *ELOVL4* cohort, but not the *PRPH2* one. The “bull’s eye like” phenotype appears to be an early stage of macular degeneration, as it corresponds in the OCT to the disruption of the EZ with a gap, which is the step preceding the collapse and complete loss of the EZ. Of note, we did not identify in our Swiss cohorts new patients affected by *PROM1*-p.(Arg373Cys)-linked “bull’s eye like” macular dystrophy (STGD4) [37].

## 4. Discussion

The purpose of our study was to determine whether clinical symptoms, onset of disease, family history, fundus autofluorescence and/or OCT findings were sufficient to discriminate among our patients between autosomal recessive *ABCA4*-associated STGD1, autosomal dominant c.810C>G p.(Tyr270*) *ELOVL4*-associated STGD3 and the autosomal dominant c.422A>G p.(Tyr141Cys) *PRPH2*-associated Stargardt-like macular dystrophy. Clinical diagnosis is difficult because these disorders linked to different genes lead to the same clinical phenotypes. Incomplete penetrance can furthermore mask a dominant inheritance in some isolated cases.

The main clinical symptom of progressive vision diminution was reported in 89% of *ABCA4* patients and in 100% of symptomatic *ELOVL4* and *PRPH2* patients. Patients of the *ABCA4* and *ELOVL4* cohorts also exhibited other classical symptoms reported in Stargardt disease: dyschromatopsia, central scotoma, photophobia, and delayed dark adaptation [1,2]. Reading difficulties are a direct consequence of these symptoms and are sometimes missed if not especially asked about. Patients of the *PRPH2* cohort did not report photophobia and dyschromatopsia, but this is very likely due to the small size of this cohort. Indeed, only a minority of patients of the *ABCA4* and *ELOVL4* cohorts reported photophobia, respectively 44% and 29%, and dyschromatopsia, respectively 11.11% and 14.29%.

The onset and progression of macular dystrophies can be objectively evaluated by assessing visual acuity (VA), despite some variability due to tiredness, ocular dryness, or presence of cataract [38,39]. We observed the typical “juvenile-onset” and “late-onset” patients in the *ABCA4* cohort (onset 10-57 years old; median 35.5 years old) [1,4,14]. In the *ELOVL4* cohort too (onset 10-79 years old; median 27 years old), we observed a rapid vision worsening among the youngest patients (12/II.2, 13/III.2 and 15/II.3) and a more conserved VA among the oldest ones (13/I.1, 13/II.1 and 14/II.4). The onset of clinical symptoms in *PRPH2*-associated autosomal dominant Stargardt-like macular dystrophy was reported to start later in life (≥50 years old) [40], which could have allowed to distinguish them from the *ABCA4* and *ELOVL4* patients. Nevertheless, the onset of clinical symptoms occurred in our small *PRPH2* cohort during childhood in two out of three patients and at age 36 for the third patient. Of note, OCT imaging was not useful for the differential diagnosis, as almost all patients showed variable extents of outer nuclear layer (ONL) atrophy and ellipsoid zone (EZ) disruption (with a gap) or EZ loss.

In the *ABCA4* cohort, five patients were affected by “juvenile-onset” STGD1 and 6 patients by “late-onset” STGD1. Pathogenic variants associated with “juvenile-onset” included the complex p.[(Leu541Pro;Ala1038Val)] (family 7, patient II.1) and the canonical splice variants c.1760+2T>C (family 2, patient II.1) and c.5018+2T>C (family 8, patient III.2). The hypomorphic variant c.5882G>A p.(Gly1961Glu) was also associated with “juvenile-onset” STGD1 in patient II.2 of family 4, with the additional variants c.3322C>T p.(Arg1108Cys) and c.6320G>A p.(Arg2107His). The hypomorphic variant c.5882G>A p.(Gly1961Glu) was present in three additional patients affected by “late-onset” STGD1. The second pathogenic *ABCA4* variant in these compound heterozygous patients was respectively c.634C>T p.(Arg212Cys) (family 6, patient II.2), c.1804C>T p.(Arg602Trp) (family 9, patient II.2) and c.3758C>T p.(Thr1253Met) (family 5, patient II.2). This last patient had been initially diagnosed with a “slow progressing retinitis pigmentosa” and still had a visual acuity of 1.0 at the age of 70. Genetic analysis did not identify any pathogenic variant in any other gene associated with retinal dystrophy, except *ABCA4*. Whether this atypical clinical phenotype is due to the presence of the second pathogenic c.3758C>T p.(Thr1253Met) variant remains elusive, because, to our best knowledge, no other identical compound heterozygous patient has been identified so far. Visual acuity of 0.8-1.0 was also relatively preserved in patient I.2 of family 10. The milder clinical phenotype may be linked to the presence of the c.4397G>A p.(Val1433Ile) variant, but the clinical symptoms could be aggravated by the frequent hypomorphic c.5603C>T p.(Asn1868Ile) variant also present in this patient. 

Family history should theoretically allow for discrimination between the autosomal recessive STGD1 and the dominantly inherited macular dystrophies. However, the *ABCA4* cohort includes family 11 with a pseudo-dominant inheritance pattern. Furthermore, patient 14/II.4, although heterozygous for the autosomal dominant *ELOVL4* variant, is the only one affected in his family and exhibits a family tree that would be compatible with an autosomal recessive inheritance pattern. This is consistent with the reported incomplete penetrance and variable expressivity in the presence of pathogenic *ELOVL4* variants [19,41]. Similarly, patient 18/II.2 is a heterozygous carrier of the autosomal dominant *PRPH2* variant, but the only one symptomatic in his family. Again, *PRPH2*-associated autosomal dominant macular dystrophies are known to have highly heterogeneous phenotypes, variable expressivity, and incomplete penetrance [40,42]. Moreover, the *PRPH2* p.(Tyr141Cys) variant can cause choroidal neovascularization in older patients [20]. This clinical phenotype at this age can easily be confused with an exudative age-related macular degeneration (AMD). This was actually the case for the mother of patient II.1 in family 17. Because of these many confounders, incomplete penetrance, variable expressivity, unknown family, adoption, consanguinity, early deaths, missed or false diagnosis, the inheritance pattern alone is not sufficient to distinguish between *ABCA4*-, *ELOVL4*- and *PRPH2*-linked macular dystrophies.

We did not perform quantitative fundus autofluorescence (qAF) in our cohorts [43]. qAF was shown to be higher in patients harboring pathogenic *ABCA4* variants, thus giving an interesting diagnostic tool for STGD1 [44]. However, high qAF levels were also observed in *PRPH2*-associated autosomal dominant Stargardt-like macular dystrophy, limiting its clinical relevance in the differential diagnosis of Stargardt and Stargardt-like macular dystrophies [45].

## 5. Conclusions

Clinical diagnosis is not sufficient to discriminate between patients affected by autosomal recessive *ABCA4*-associated STGD1, autosomal dominant c.810C>G p.(Tyr270*) *ELOVL4*-associated STGD3 and the autosomal dominant c.422A>G p.(Tyr141Cys) *PRPH2*-associated Stargardt-like macular dystrophy. Even a small population sample of fewer than 8 million people like the Swiss population harbors all the diversity of Stargardt and Stargardt-like phenotypes. Therefore, genetic testing is required for all these patients. The c.810C>G p.(Tyr270*) variant in the *ELOVL4* should be suspected and specifically tested for in all patients originating from that small mountain region in the Southern part of the Canton of Valais. A clear molecular diagnosis in Stargardt and Stargardt-like patients also has some direct benefits for patients. First, genetically confirmed STGD1 patients should stop any vitamin A supplementation to avoid an increase in lipofuscin deposition in the RPE [46]. Second, follow-up controls in patients affected by c.422A>G p.(Tyr141Cys) *PRPH2*-associated Stargardt-like macular dystrophy will carefully evaluate the risk to develop choroidal neovascularization, which could lead to hemorrhages and subsequent loss of their residual visual acuity [20]. When appropriate, these patients could then benefit from anti-VEGF intravitreal injections. Third, the molecular diagnosis is necessary to establish autosomal recessive or autosomal dominant inheritance, a prerequisite for family planning and risk calculations. Finally, even though no treatment currently exist for these three macular dystrophies, when therapy will be available, only patients with a confirmed molecular diagnosis will benefit from it.

## Figures and Tables

**Figure 1 genes-12-00812-f001:**
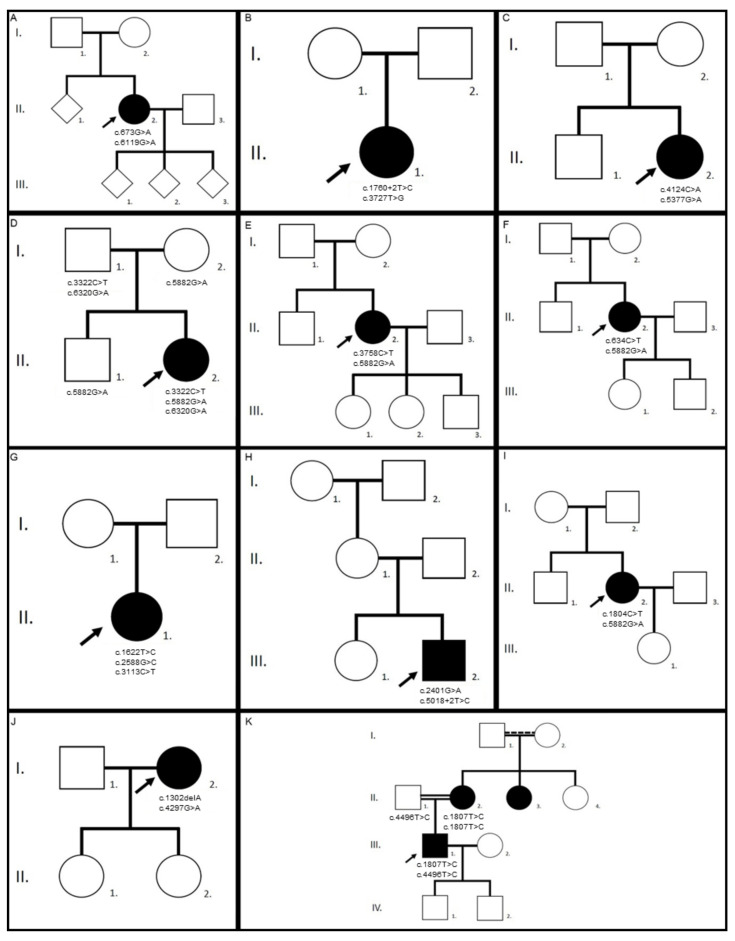
Family trees of the *ABCA4* cohort. In each pedigree, the arrow indicates the proband. Family 1, patient II.2 (**A**), family 2, patient II.1 (**B**), family 3, patient II.2 (**C**), family 4, patient II.2 (**D**), family 5, patient II.2 (**E**), family 6, patient II.2 (**F**), family 7, patient II.1 (**G**), family 8, patient III.2 (**H**), family 9, patient II.2 (**I**), family 10, patient I.2 (**J**), and family 11, patient III.1 (**K**). Identified pathogenic variants are further detailed in Table 1.

**Figure 2 genes-12-00812-f002:**
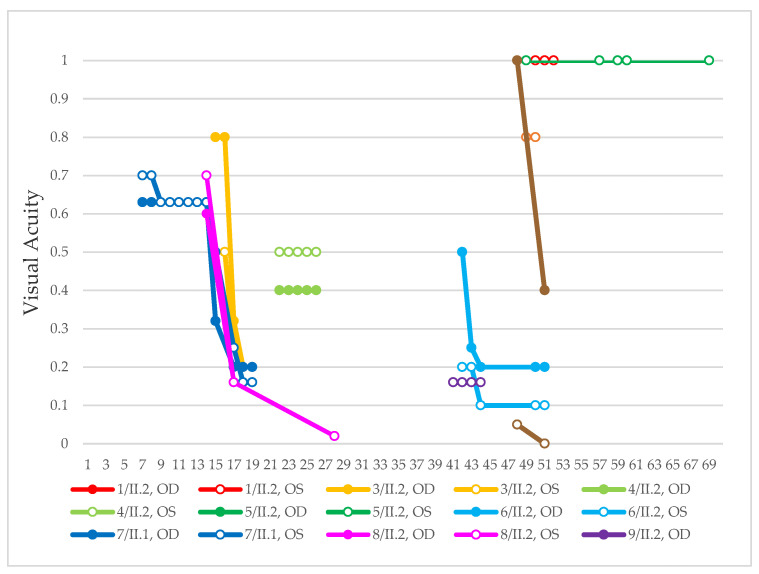
Evolution of the visual acuity in patients harboring pathogenic bi-allelic *ABCA4* variants. Best corrected visual acuity was assessed at indicated age (years), each dot indicating a clinical examination. OD, right eye; OS, left eye. Patient descriptions in text and tables.

**Figure 3 genes-12-00812-f003:**
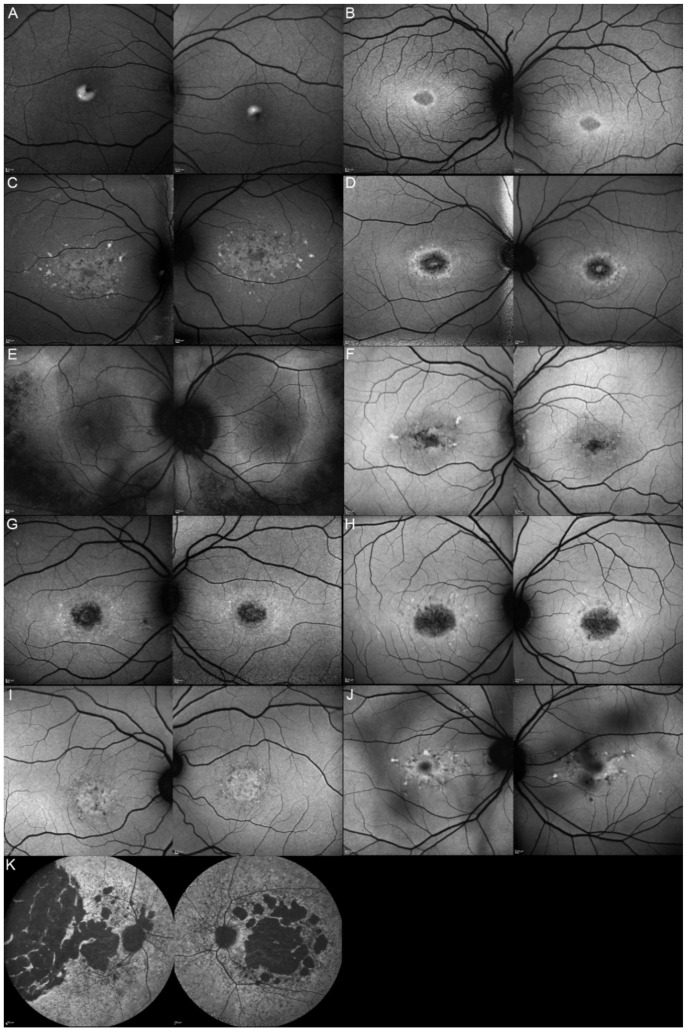
Fundus autofluorescence (FAF) imaging of patients harboring pathogenic bi-allelic *ABCA4* variants. Patient 1/II.2 (**A**), patient 2/II.1 (**B**), patient 3/II.2 (**C**), patient 4/II.2 (**D**), patient 5/II.2 (**E**), patient 6/II.2 (**F**), patient 7/II.1 (**G**), patient 8/III.2 (**H**), patient 10/I.2 (**J**), and patient 11/III.1 (**K**). For each patient, the left panel shows the right eye, and the right panel the left one. The FAF findings are described in detail in Table 2.

**Figure 4 genes-12-00812-f004:**
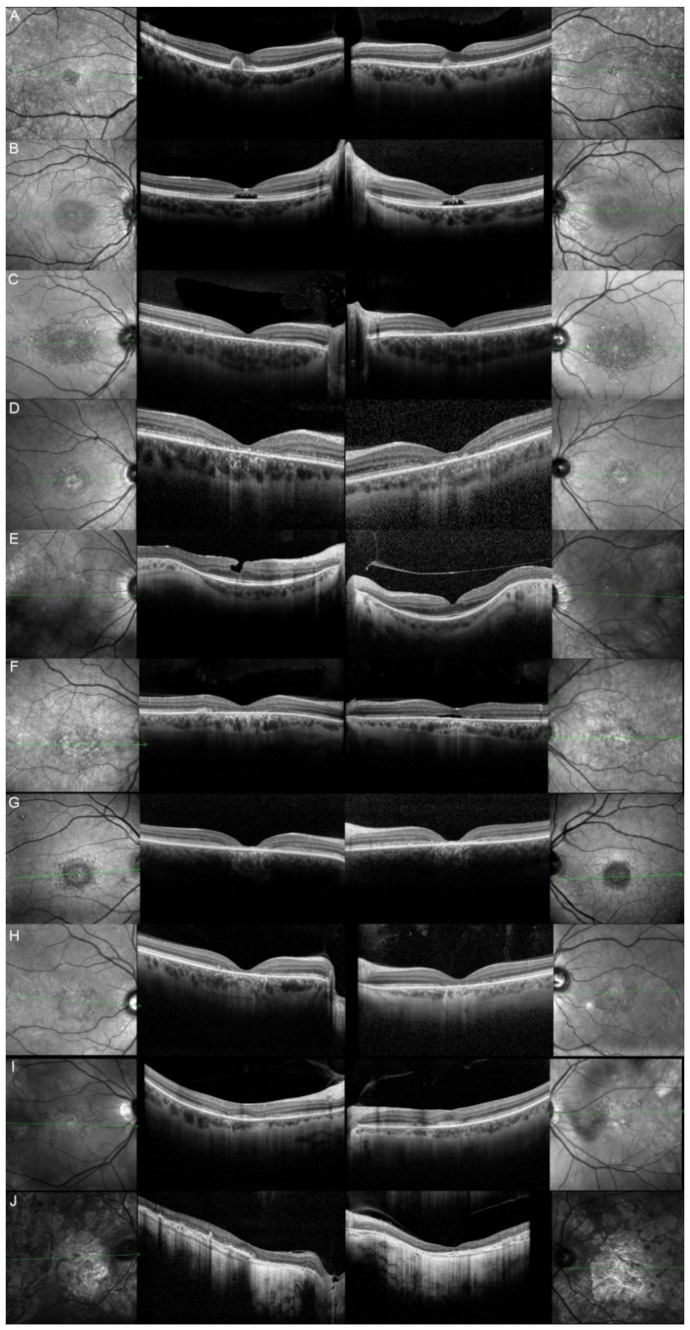
Optical coherence tomography (OCT) of patients harboring pathogenic bi-allelic *ABCA4* variants. Patient 1/II.2 (**A**), patient 2/II.1 (**B**), patient 3/II.2 (**C**), patient 4/II.2 (**D**), patient 5/II.2 (**E**), patient 6/II.2 (**F**), patient 7/II.1 (**G**), patient 9/II.2 (**H**), patient 10/I.2 (**I**), and patient 11/III.1 (**J**). For each patient, green lines on FAF images indicate the scanning sections (left and right panels) of the OCT (middle panels). The left panels show the right eye, and the right panels the left one. The OCT findings are described in detail in Table 2.

**Figure 5 genes-12-00812-f005:**
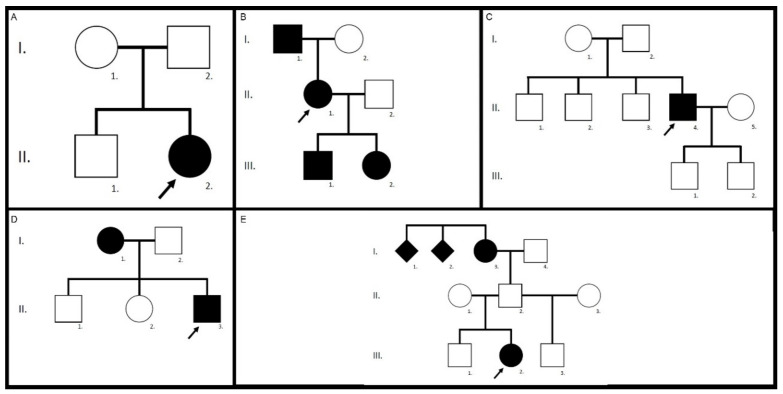
Family trees of the *ELOVL4* cohort. Family 12, patient II.2 (**A**), family 13, patients I.1, II.1, III.1 and III.2 (**B**), family 14, patient II.4 (**C**), family 15, patient II.3 (**D**), and family 16, patient III.2 (**E**).

**Figure 6 genes-12-00812-f006:**
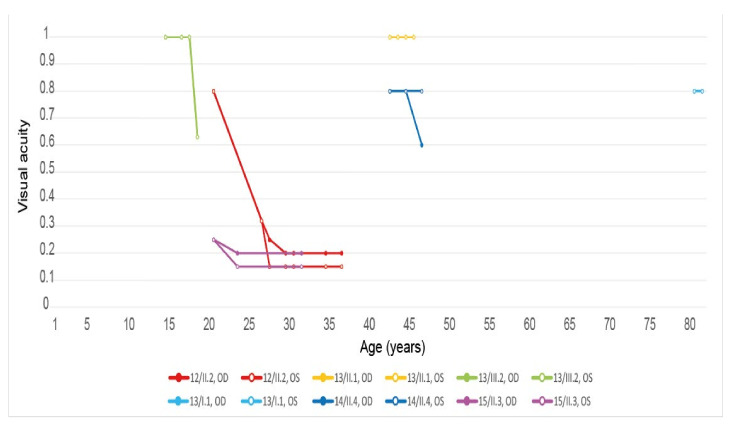
Evolution of the visual acuity in patients harboring the pathogenic c.810C>G, p.(Tyr270*) *ELOVL4* variant. Best corrected visual acuity was assessed at indicated age (years), each dot indicating a clinical examination. OD, right eye; OS, left eye. Patient descriptions in text and tables.

**Figure 7 genes-12-00812-f007:**
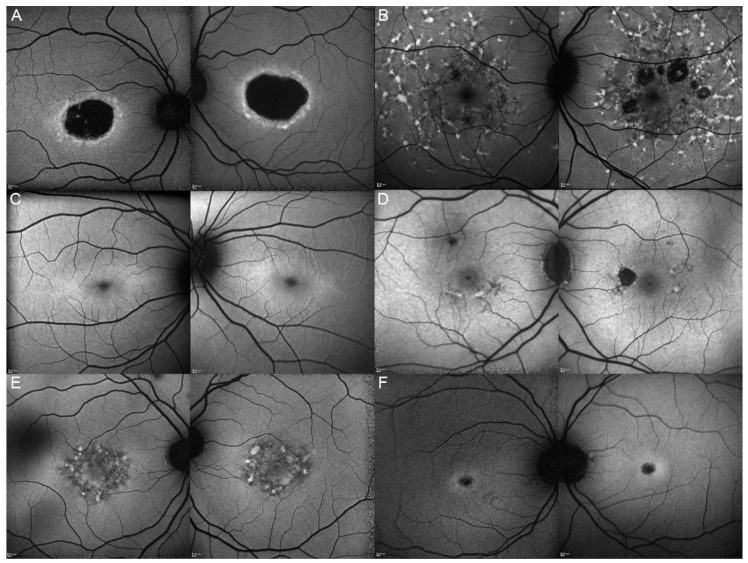
Fundus autofluorescence imaging of patients harboring the pathogenic c.810C>G p.(Tyr270*) *ELOVL4* variant. Patient 12/II.2 (**A**), patient 13/II.1 (**B**), patient 13/III.2 (**C**), patient 13/I.1 (**D**), patient 14/II.4 (**E**), and patient 15/II.3 (**F**). For each patient, the left panel shows the right eye, and the right panel the left one. The FAF findings are described in detail in Table 2.

**Figure 8 genes-12-00812-f008:**
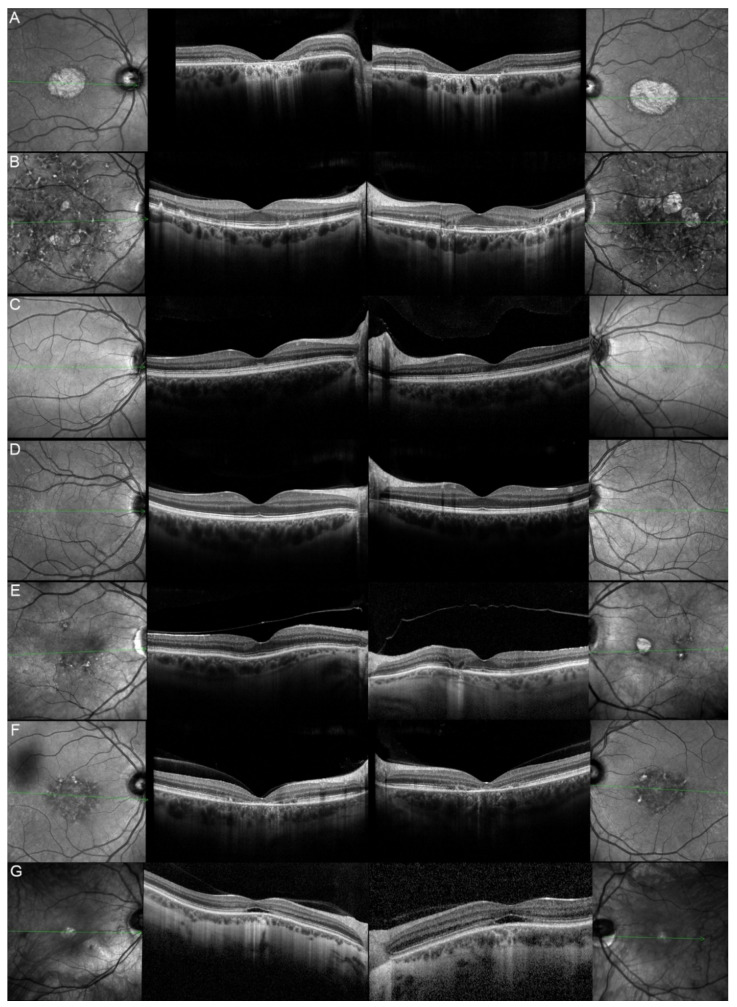
Optical coherence tomography (OCT) of patients harboring the pathogenic c.810C>G, p.(Tyr270*) *ELOVL4* variant. Patient 12/II.2 (**A**), patient 13/II.1 (**B**), patient 13/III.2 (**C**), patient 13/III.1 (**D**), patient 13/I.1 (**E**), patient 14/II.4 (**F**), and patient 15/II.3 (**G**). For each patient, green lines on FAF images indicate the scanning sections (left and right panels) of the OCT (middle panels). The left panels show the right eye, and the right panels the left one. The OCT findings are described in detail in Table 2.

**Figure 9 genes-12-00812-f009:**
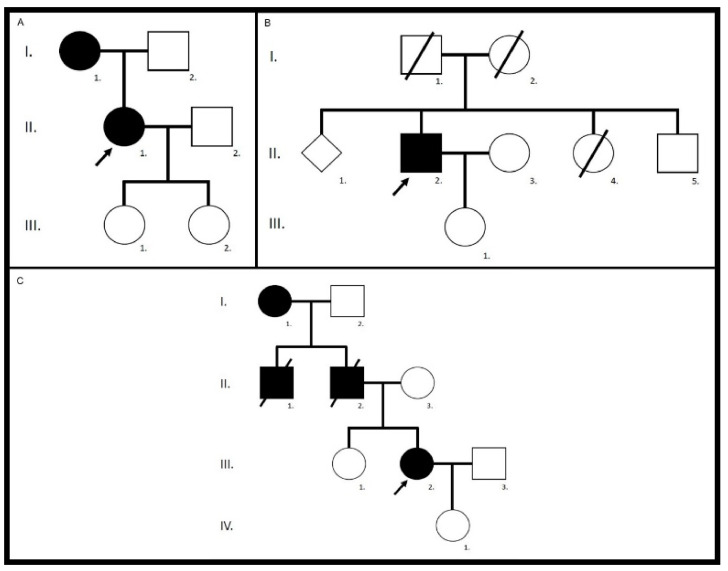
Family trees of the *PRPH2* cohort. Family 17, patient II.1 (**A**), family 18, patient II.2 (**B**), and family 19, patient III.2 (**C**).

**Figure 10 genes-12-00812-f010:**
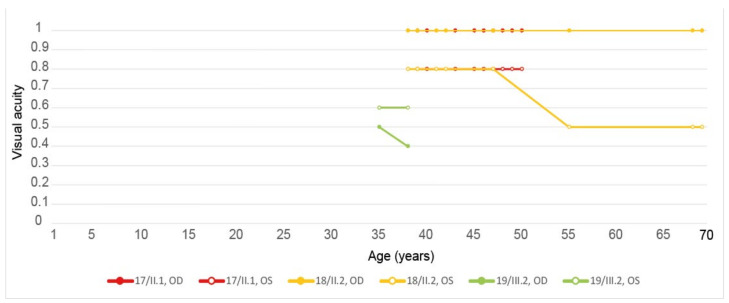
Evolution of the visual acuity in patients harboring the pathogenic c.422A>G p.(Tyr141Cys) *PRPH2* variant. Best corrected visual acuity was assessed at indicated age (years), each dot indicating a clinical examination. OD, right eye; OS, left eye. Patient descriptions in text and tables.

**Figure 11 genes-12-00812-f011:**
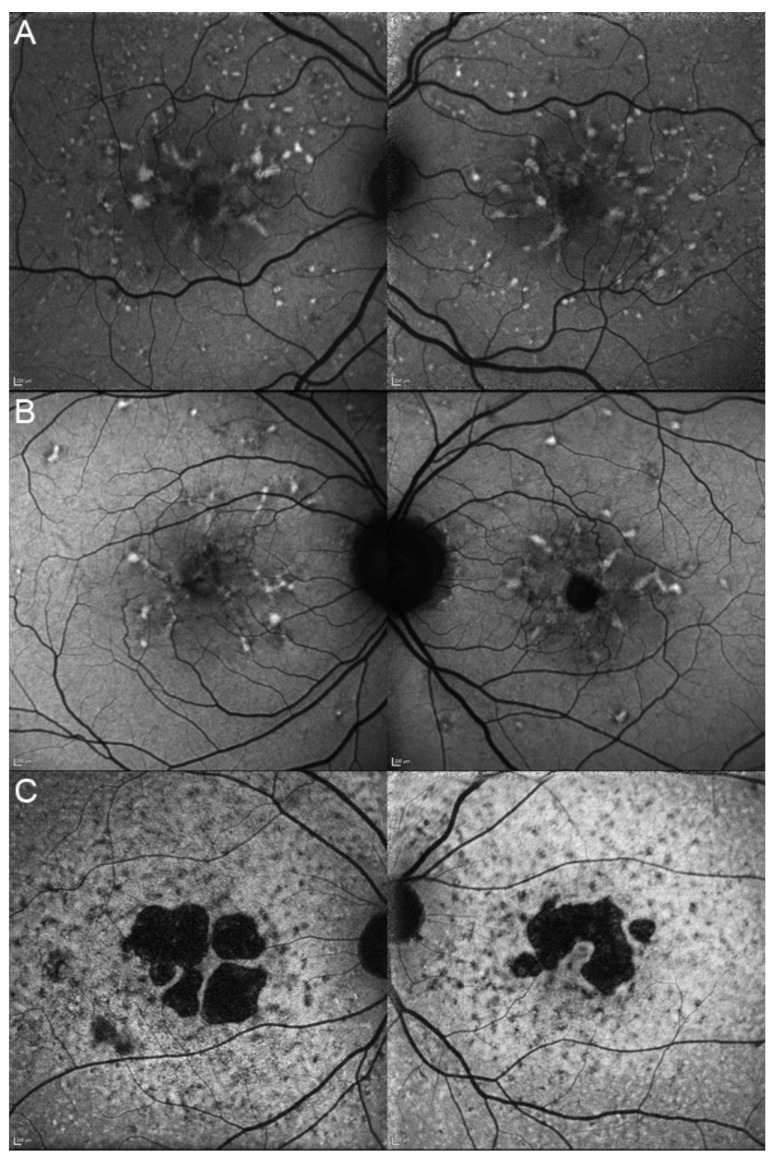
Fundus autofluorescence imaging of patients harboring the pathogenic c.422A>G p.(Tyr141Cys) *PRPH2* variant. Patient 17/II.1 (**A**), patient 18/II.2 (**B**), patient 19/III.2 (**C**). For each patient, the left panel shows the right eye, and the right panel the left one. The FAF findings are described in detail in Table 2.

**Figure 12 genes-12-00812-f012:**
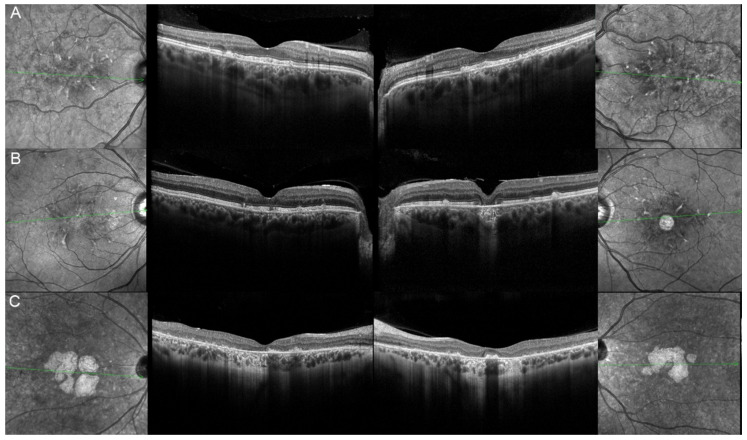
Optical coherence tomography (OCT) of patients harboring the pathogenic c.422A>G p.(Tyr141Cys) *PRPH2* variant. Patient 17/II.1 (**A**), patient 18/II.2 (**B**), patient 19/III.2 (**C**). For each patient, green lines on FAF images indicate the scanning sections (left and right panels) of the OCT (middle panels). The left panels show the right eye, and the right panels the left one. The OCT findings are described in detail in Table 2.

**Figure 13 genes-12-00812-f013:**
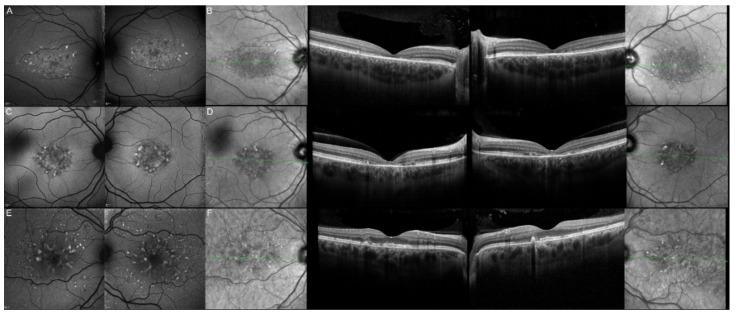
“Flecks” clinical phenotype. FAF (**A**) and OCT (**B**) imaging of STGD1 patient 3/II.2 compound heterozygous for the c.4124C>A p.(Ala1375Glu) and c.5377G>A p.(Val1793Met) *ABCA4* variants. FAF (**C**) and OCT (**D**) imaging of STGD3 patient 14/II.4 harboring the pathogenic c.810C>G p.(Tyr270*) *ELOVL4* variant. FAF (**E**) and OCT (**F**) imaging of patient 17/II.1 affected by c.422A>G p.(Tyr141Cys) *PRPH2*-associated autosomal dominant Stargardt-like macular dystrophy.

**Figure 14 genes-12-00812-f014:**
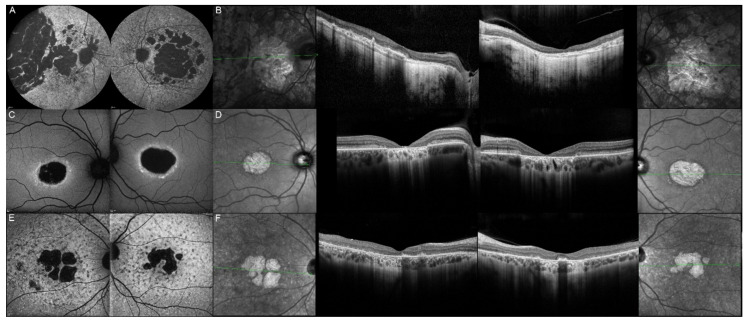
“Atrophy” clinical phenotype. FAF (**A**) and OCT (**B**) imaging of STGD1 patient 11/III.1 compound heterozygous for the c.1807T>C p.(Tyr603His) and c.4496T>C p.(Leu1499Pro) *ABCA4* variants. FAF (**C**) and OCT (**D**) imaging of STGD3 patient 12/II.2 harboring the pathogenic c.810C>G p.(Tyr270*) *ELOVL4* variant. FAF (**E**) and OCT (**F**) imaging of patient 19/III.2 affected by c.422A>G p.(Tyr141Cys) *PRPH2*-associated autosomal dominant Stargardt-like macular dystrophy.

**Figure 15 genes-12-00812-f015:**
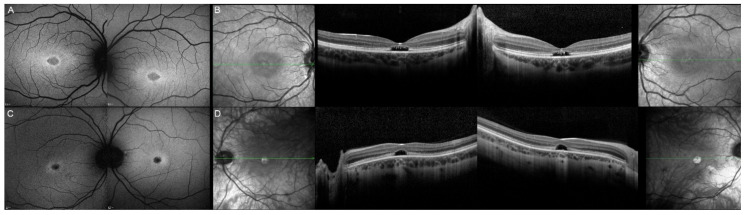
“Bull’s eye like” clinical phenotype. FAF (**A**) and OCT (**B**) imaging of STGD1 patient 2/II.1 compound heterozygous for the c.1760+2T>C p.? and c.3727T>G p.(Tyr1243Asp) *ABCA4* variants. FAF (**C**) and OCT (**D**) imaging of STGD3 patient 12/II.2 harboring the pathogenic c.810C>G p.(Tyr270*) *ELOVL4* variant.

**Table 1 genes-12-00812-t001:** Patients’ genotypes. The reference transcripts for variant numbering using A in ATG as number 1 are NM_000350.3 for *ABCA4*, NM_022726.3 for *ELOVL4,* and NM_000322.5 for *PRPH2*. Pathogenicity is according to the criteria of the American College of Medical Genetics and Genomics (ACMG) [24]. VUS, variant of unknown significance.

Family/ patient	Gene	Variants	Zygosity	Pathogenicity	First Description
**1/II.2**	*ABCA4*	c.673G>A p.(Val225Met)	heterozygous	likely pathogenic	[25]
		c.6119G>A p.(Arg2040Gln)	heterozygous	pathogenic	[4]
**2/II.1**	*ABCA4*	c.1760+2T>C p.?	heterozygous	pathogenic	novel
		c.3727T>G p.(Tyr1243Asp)	heterozygous	likely pathogenic	[25]
**3/II.2**	*ABCA4*	c.4124C>A p.(Ala1375Glu)	heterozygous	likely pathogenic	[26]
		c.5377G>A p.(Val1793Met)	heterozygous	likely pathogenic	[26]
**4/II.2**	*ABCA4*	c.3322C>T p.(Arg1108Cys)	heterozygous	pathogenic	[27]
		c.5882G>A p.(Gly1961Glu)	heterozygous	pathogenic	[27]
		c.6320G>A p.(Arg2107His)	heterozygous	VUS	[28]
**5/II.2**	*ABCA4*	c.3758C>T p.(Thr1253Met)	heterozygous	likely pathogenic	[29]
		c.5882G>A p.(Gly1961Glu)	heterozygous	pathogenic	[27]
**6/II.2**	*ABCA4*	c.634C>T p.(Arg212Cys)	heterozygous	pathogenic	[30]
		c.5882G>A p.(Gly1961Glu)	heterozygous	pathogenic	[27]
**7/II.1**	*ABCA4*	c.1622T>C p.(Leu541Pro)	heterozygous	likely pathogenic	[27]
		c.2588G>C p.[(Gly863Ala,Gly863del)]	heterozygous	likely pathogenic	[31]
		c.3113C>T p.(Ala1038Val)	heterozygous	pathogenic	[27]
**8/III.2**	*ABCA4*	c.2401G>A p.(Ala801Thr)	heterozygous	likely pathogenic	[32]
		c.5018+2T>C p.?	heterozygous	pathogenic	[33]
**9/II.2**	*ABCA4*	c.1804C>T p.(Arg602Trp)	heterozygous	pathogenic	[27]
		c.5882G>A p.(Gly1961Glu)	heterozygous	pathogenic	[27]
**10/I.2**	*ABCA4*	c.1302delA p.(Gln437Arg*fs**12)	heterozygous	pathogenic	[34]
		c.4297G>A p.(Val1433Ile)	heterozygous	likely pathogenic	[35]
**11/III.1**	*ABCA4*	c.1807T>C p.(Tyr603His)	heterozygous	likely pathogenic	[36]
		c.4496T>C p.(Leu1499Pro)	heterozygous	likely pathogenic	novel
**12/II.2**	*ELOVL4*	c.810C>G p.(Tyr270*)	heterozygous	pathogenic	[13]
**13/II.1**	*ELOVL4*	c.810C>G p.(Tyr270*)	heterozygous	pathogenic	[13]
**13/III.2**	*ELOVL4*	c.810C>G p.(Tyr270*)	heterozygous	pathogenic	[13]
**13/III.1**	*ELOVL4*	c.810C>G p.(Tyr270*)	heterozygous	pathogenic	[13]
**13/I.1**	*ELOVL4*	c.810C>G p.(Tyr270*)	heterozygous	pathogenic	[13]
**14/II.4**	*ELOVL4*	c.810C>G, p.(Tyr270*)	heterozygous	pathogenic	[13]
**15/II.3**	*ELOVL4*	c.810C>G p.(Tyr270*)	heterozygous	pathogenic	[13]
**16/III.2**	*ELOVL4*	c.810C>G p.(Tyr270*)	heterozygous	pathogenic	[13]
**17/II.1**	*PRPH2*	c.422A>G p.(Tyr141Cys)	heterozygous	pathogenic	[16]
**18/II.2**	*PRPH2*	c.422A>G p.(Tyr141Cys)	heterozygous	pathogenic	[16]
**19/III.2**	*PRPH2*	c.422A>G p.(Tyr141Cys)	heterozygous	pathogenic	[16]

**Table 2 genes-12-00812-t002:** Summary of fundus autofluorescence (FAF) and optical coherence tomography (OCT) findings. EZ, ellipsoid zone; FAF, fundus autofluorescence; HOAF, hypoautofluorescence; HPAF, hyperautofluorescence; NA, not available; OD: right eye; ONL, outer nuclear layer; OS, left eye.

Patient	Gene	Age	Sex	FAF	OCT
**1, II.2**	ABCA4	53	F	Foveal HPAF	Subretinal hyperreflective deposits
**2, II.1**	ABCA4	18	F	Central HOAF; concentric ring of HPAF	Central EZ disruption with gap; ONL atrophy
**3, II.2**	ABCA4	18	F	Foveal HOAF; central mottled pattern of HPAF and HOAF flecks	Central EZ loss; central ONL atrophy
**4, II.2**	ABCA4	27	F	Foveal HPAF; parafoveal HOAF; paramacular HPAF	Central EZ loss; central ONL atrophy
**5, II.2**	ABCA4	69	F	Mid-peripheral mottled HOAF	Mid-peripheral ONL atrophy, lamellar macular hole (OD)
**6, II.2**	ABCA4	53	F	Foveal HOAF; central mottled pattern of HPAF and HOAF flecks	Central EZ disruption with gap (OS) and loss (OD); central ONL atrophy
**7, II.1**	ABCA4	19	F	Central HOAF; HPAF flecks	Central EZ loss; central ONL atrophy
**8, III.2**	ABCA4	30	M	Central HOAF; HPAF flecks	NA
**9, II.2**	ABCA4	44	F	Central mottled pattern of HPAF and HOAF flecks	Central EZ loss; central ONL atrophy
**10, I.2**	ABCA4	51	F	Foveal HOAF; central mottled pattern of HPAF and HOAF flecks	Perifoveal EZ loss; perifoveal ONL atrophy; loss of the foveal depression
**11, III.1**	ABCA4	53	M	Extended central and peripheral HOAF; panretinal diffuse mottled pattern of HPAF and HOAF flecks	ONL atrophy; peripheral choroidal atrophy; RPE atrophy; subretinal hyperreflective deposits
**12, II.2**	ELOVL4	37	F	Central HOAF; concentric ring of HPAF	Central EZ loss; central ONL atrophy and RPE atrophy
**13, II.1**	ELOVL4	46	F	Multifocal areolar parafoveal HOAF; mottled pattern of HPAF and HOAF flecks; perimacular to mid-peripheral HPAF flecks	Perimacular EZ loss; perimacular ONL atrophy; subretinal hyperreflective deposits
**13, III.2**	ELOVL4	19	F	No special features	Subfoveal hyperreflective deposits (OD)
**13, III.1**	ELOVL4	21	M	NA	No special features
**13, I.1**	ELOVL4	82	M	Areolar HOAF; paracentral mottled pattern of HPAF and HOAF flecks	ONL thinning; perimacular localised EZ loss (OS)
**14, II.4**	ELOVL4	48	M	Central mottled pattern of HPAF and HOAF flecks	Central EZ loss; central ONL atrophy; perimacular subretinal hyperreflective deposits
**15, II.3**	ELOVL4	34	M	Subfoveal HOAF; concentric ring of HPAF	Central EZ disruption with gap; central ONL atrophy
**16, III.2**	ELOVL4	28	F	NA	NA
**17, II.1**	PRPH2	51	F	Butterfly pattern of HPAF and HOAF flecks; paramacular to mid-peripheral HPAF flecks	Irregular EZ thickening
**18, II.2**	PRPH2	69	M	Subfoveal HOAF (OS); butterfly pattern of HPAF and HOAF flecks; paramacular to mid-peripheral HPAF flecks	Central EZ loss (OS); central ONL atrophy (OS); localised ONL thinning (OD)
**19, III.2**	PRPH2	39	F	Central areolar HOAF; disseminated nummular HPAF	Extended retinal atrophy

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
