# Peer review of "Absence of Genotype/Phenotype Correlations Requires Molecular Diagnostic to Ascertain Stargardt and Stargardt-Like Swiss Patients"

_genes, 2021, doi:10.3390/genes12060812_

Round 1

Reviewer 1 Report

Dear Editor,

The authors genetically analysed 22 Swiss patients diagnosed with Stargardt disease. They identified variants in ABCA4, ELOVL4 and PRPH2 genes. They highlight that clinical diagnosis is not sufficient to discriminate between patients affected by Stargardt-like phenotypes and that genotyping is mandatory to differentiate them and to confirm the diagnosis.

Please find below my comments and suggestions.

line 53 : "there" to suppress

line 66 : "variants in non-coding regions, and deep intronic variants": this is redundant as deep intronic variants are non-coding variants. Maybe replace by non-canonical splice site variants and deep intronic variants.

line 73: Not true, not always a much more severe form because the p.(Gly1961Glu) is an hypomorphic variant and even at the compound heterozygous state, milder phenotype can occur (Cornelis et al 2017). see the "D-E-F-I" phenotypes and loss of visual acuity.

line 74: Please turn it differently "Similar phenotypes to that present in STGD1 patients can also be observed in...". Clinical diagnosis is difficult because different gene-associated disorders lead to same phenotypes. More, some isolated cases, due to incomplete penetrance can mask a dominant inheritance.

line 97: The authors should try to better introduce the aim of the manuscript. "Given the genetic heterogeneity of patients presenting a Stargardt-like phenotype, we aimed at deciphering a genotype-phenotype correlation through the study of a cohort of Swiss patients. ". Stargardt (STGD1) is due to ABCA4 variants, so Stargardt disease phenotype should not be apply for phenotypes associated with other genes. Maybe it is more adapted to use the term STGD-like.

line 114: The authors should clearly precise in the text that exome sequencing has been done

line 117: The frequent hypomorphic variant c.5603C>T could not be called with a filter above 5%. This variant is frequently in cis with the c.2588G>C and makes it fully penetrant. Can you precise if this variant was checked and was present in some patients?

figure 1: Try to be consistent with the annotation, used either c. or p. annotation, or both, but not one or the other one. When segregation was confirmed (D, K), please, use the correct nomenclature according to HGVS, rather than exon x, exon y.... For (G) : c.2588G>C, p.[(Gly863Ala,Gly863del)] like in table 1; For (G): p.[(Leu541Pro;Ala1038Val)] is a frequent complex allele.

-family E seems to be doubtful according to the autofluorescence, the late onset, the absence of BAV. As the segregation has not been done, can you confirm and justify the Stargardt diagnosis?

- Can the authors highlight the importance to check the ELOVL4 variant in patient originating from this small area in Swiss as this is a founder mutation? Can the authors provide the frequency of the variant in gnomAD? It is a quite rare variant in other European countries.

- line 277: 2 times "reported"

- I really appreciate the comparison of the three phenotypes, flecks, BEM and atrophy. However, it seems to be more results than discussion

- family 7: is the p.(Asn1868Ile) present?

- family 10: to me, the p.(Val1433Ile) variant is doubtful and likely benign according to ACMG 2015. Can the authors precise in the manuscript that some frequent non-canonical splice site variants cannot be found by the exome analysis, like c.5461-10T>C or c.4253+43G>A for instance. It cannot be ruled out that other more pathogenic variants (non-canonical or deep intronic variants) can be present in cis of the p.(Val1433Ile). Have the copy number variant been assessed? The authors should explain this point and justify the STGD1 phenotype.

 - families 4 5 6 9: each carry the Gly1961Glu. Can you detailed if they present the same phenotypes? what about the age at onset for the loss of visual acuity?

- The authors should try to better correlate the visual acuity follow up with the ABCA4 variants type. They should talk about the hypomorphic variants associated with milder or later-onset forms. Overwise, the manuscript is clear, well written and it brings an interesting comparison of same presentations for different genes.

Author Response

Reviewer 1

We are very grateful to Reviewer 1 for the very competent and detailed review with many insightful and constructive comments. Below our point-per-point responses.

line 53 : "there" to suppress

ok, suppressed

line 66 : "variants in non-coding regions, and deep intronic variants": this is redundant as deep intronic variants are non-coding variants. Maybe replace by non-canonical splice site variants and deep intronic variants.

Thank you for this remark, we corrected the text according to your suggestion.

line 73: Not true, not always a much more severe form because the p.(Gly1961Glu) is an hypomorphic variant and even at the compound heterozygous state, milder phenotype can occur (Cornelis et al 2017). see the "D-E-F-I" phenotypes and loss of visual acuity.

Thank you very much for bringing this to our attention: we modified the text accordingly.

line 74: Please turn it differently "Similar phenotypes to that present in STGD1 patients can also be observed in...". Clinical diagnosis is difficult because different gene-associated disorders lead to same phenotypes. More, some isolated cases, due to incomplete penetrance can mask a dominant inheritance.

We have modified the sentence according to your suggestion and integrated the second part of your comment into the conclusion (1st paragraph of the discussion).

line 97: The authors should try to better introduce the aim of the manuscript. "Given the genetic heterogeneity of patients presenting a Stargardt-like phenotype, we aimed at deciphering a genotype-phenotype correlation through the study of a cohort of Swiss patients. ". Stargardt (STGD1) is due to ABCA4 variants, so Stargardt disease phenotype should not be apply for phenotypes associated with other genes. Maybe it is more adapted to use the term STGD-like.

Thank you very much for this excellent wording. We have modified the text accordingly.

line 114: The authors should clearly precise in the text that exome sequencing has been done

We have modified the text accordingly.

line 117: The frequent hypomorphic variant c.5603C>T could not be called with a filter above 5%. This variant is frequently in cis with the c.2588G>C and makes it fully penetrant. Can you precise if this variant was checked and was present in some patients?

Yes, we had checked for this variant in all patients and added in the Results section, 1st paragraph, the following note: We also checked for the presence of the frequent hypomorphic variant c.5603A>T p.(Asn1868Ile) and identified it in patient I.2 of family 10.

figure 1: Try to be consistent with the annotation, used either c. or p. annotation, or both, but not one or the other one. When segregation was confirmed (D, K), please, use the correct nomenclature according to HGVS, rather than exon x, exon y.... For (G) : c.2588G>C, p.[(Gly863Ala,Gly863del)] like in table 1;

We have modified the nomenclature in Figure 1, and for space reasons opted for the ‘c.’ annotation only.

For (G): p.[(Leu541Pro;Ala1038Val)] is a frequent complex allele.

Yes, we are aware of this and have added a note in the text (first paragraph Results).

Family E seems to be doubtful according to the autofluorescence, the late onset, the absence of BAV. As the segregation has not been done, can you confirm and justify the Stargardt diagnosis?

This patient has been initially diagnosed with a slow-progressing RP. No pathogenic variants, except for ABCA4 were identified. We discuss now this patient together with the other ones harboring the p.(Gly1961Glu) variant (see below). Of note, the parents of the patient had already died at respective ages of 92 and 94 when she was referred by her primary practitioner to our clinic. The patient’s parents had no history of retinal dystrophy.

Can the authors highlight the importance to check the ELOVL4 variant in patient originating from this small area in Swiss as this is a founder mutation? Can the authors provide the frequency of the variant in gnomAD? It is a quite rare variant in other European countries.

We have modified the abstract accordingly. There is no reported frequency of this variant in gnomAD! We also added detailed comments to the Conclusions sections with respect to genetic testing.

line 277: 2 times "reported"

We replaced the first ‘reported’ with ‘exhibited’.

I really appreciate the comparison of the three phenotypes, flecks, BEM and atrophy. However, it seems to be more results than discussion

We have moved the three figures 13,14 and 15 together with the accompanying text into results.

Family 7: is the p.(Asn1868Ile) present?

No the c.5603C>T p.(Asn1868Ile) variant is not present.

Family 10: to me, the p.(Val1433Ile) variant is doubtful and likely benign according to ACMG 2015. Can the authors precise in the manuscript that some frequent non-canonical splice site variants cannot be found by the exome analysis, like c.5461-10T>C or c.4253+43G>A for instance. It cannot be ruled out that other more pathogenic variants (non-canonical or deep intronic variants) can be present in cis of the p.(Val1433Ile). Have the copy number variant been assessed? The authors should explain this point and justify the STGD1 phenotype.

We have rechecked classification according to the ACMG 2015 criteria, and, again, it is classified as a likely pathogenic variant. We discuss now this patient separately in the Discussion, mentioning also the presence of the c.5603C>T p.(Asn1868Ile) variant that may contribute to the clinical phenotype.

Families 4 5 6 9: each carry the Gly1961Glu. Can you detailed if they present the same phenotypes? what about the age at onset for the loss of visual acuity?

We have no added a paragraph in the discussion about this:

In the ABCA4 cohort, five patients were affected by ‘juvenile-onset’ STGD1 and 6 patients by ‘late-onset’ STGD1. Pathogenic variants associated with ‘juvenile-onset’ included the complex p.[(Leu541Pro;Ala1038Val)] (family 7, patient II.1) and the canonical splice variants c.1760+2T>C (family 2, patient II.1) and c.5018+2T>C (family 8, patient III.2). The hypomorphic variant c.5882G>A p.(Gly1961Glu) was also associated with ‘juvenile-onset’ STGD1 in patient II.2 of family 4, with the additional variants c.3322C>T p.(Arg1108Cys) and c.6320G>A p.(Arg2107His). The hypomorphic variant c.5882G>A p.(Gly1961Glu) was present in three additional patients affected by ‘late-onset’ STGD1. The second pathogenic ABCA4 variant in these compound hetero-zygous patients was respectively c.634C>T p.(Arg212Cys) (family 6, patient II.2), c.1804C>T p.(Arg602Trp) (family 9, patient II.2) and c.3758C>T p.(Thr1253Met) (family 5, patient II.2). This last patient had been initially diagnosed with a ‘slow progressing retinitis pigmentosa’ and still had a visual acuity of 1.0 at the age of 70. Genetic analy-sis did not identify any pathogenic variant in any other gene associated with retinal dystrophy, except ABCA4. Whether this atypical clinical phenotype is due to the pres-ence of the second pathogenic c.3758C>T p.(Thr1253Met) variant remains elusive, be-cause, to our best knowledge, no other identical compound heterozygous patient has been identified so far. Visual acuity of 0.8-1.0 was also relatively preserved in patient I.2 of family 10. The milder clinical phenotype may be linked to the presence of the c.4397G>A p.(Val1433Ile) variant, but the clinical symptoms could be aggravated by the frequent hypomorphic c.5603C>T p.(Asn1868Ile) variant also present in this patient.

The authors should try to better correlate the visual acuity follow up with the ABCA4 variants type. They should talk about the hypomorphic variants associated with milder or later-onset forms.

As mentioned above, we have added such a paragraph to the discussion.

Reviewer 2 Report

In the manuscript “Absence of genotype/phenotype correlations requires molecular diagnostic to ascertain Stargardt and Stargardt-like Swiss patients” by Buhler VMM et al., the authors present detailed clinical data, including progression of visual acuity, clinical symptoms, inheritance patterns, fundus autofluorescence and optical coherence tomography of 22 patients with a genetically confirmed cause of macular dystrophy. Their observations are in line with previous findings on a broad phenotypical spectrum of Stargardt disease and provide another convincing piece of evidence showing that clinical diagnosis is not sufficient to suspect a particular causative gene and genetic testing is required for all of the patients. The manuscript is well-written and the text is supported by tables and figures nicely demonstrating the reported phenotypes.

Please include labels for the X-axis in the Figure 2, 6 and 10.

Author Response

Reviewer 2

In the manuscript “Absence of genotype/phenotype correlations requires molecular diagnostic to ascertain Stargardt and Stargardt-like Swiss patients” by Buhler VMM et al., the authors present detailed clinical data, including progression of visual acuity, clinical symptoms, inheritance patterns, fundus autofluorescence and optical coherence tomography of 22 patients with a genetically confirmed cause of macular dystrophy. Their observations are in line with previous findings on a broad phenotypical spectrum of Stargardt disease and provide another convincing piece of evidence showing that clinical diagnosis is not sufficient to suspect a particular causative gene and genetic testing is required for all of the patients. The manuscript is well-written and the text is supported by tables and figures nicely demonstrating the reported phenotypes.

Please include labels for the X-axis in the Figure 2, 6 and 10.

We thank this reviewer for his positive feedback. We have added the labels for Figures 2, 6 and 10 and also improved the overall presentation of these figures.
